# Generation of Cytotoxic T Cells and Dysfunctional CD8 T Cells in Severe COVID-19 Patients

**DOI:** 10.3390/cells11213359

**Published:** 2022-10-25

**Authors:** Sarah Cristina Gozzi-Silva, Luana de Mendonça Oliveira, Ricardo Wesley Alberca, Natalli Zanete Pereira, Fábio Seiti Yoshikawa, Anna Julia Pietrobon, Tatiana Mina Yendo, Milena Mary de Souza Andrade, Yasmim Alefe Leuzzi Ramos, Cyro Alves Brito, Emily Araujo Oliveira, Danielle Rosa Beserra, Raquel Leão Orfali, Valéria Aoki, Alberto Jose da Silva Duarte, Maria Notomi Sato

**Affiliations:** 1Institute of Tropical Medicine, University of São Paulo, São Paulo 05403-000, Brazil; 2Department of Immunology, Institute of Biomedical Sciences, University of São Paulo, São Paulo 05508-000, Brazil; 3Laboratory of Dermatology and Immunodeficiencies 56 (LIM-56), Division of Dermatology, Medical School, University of São Paulo, Av. Dr. Enéas Carvalho de Aguiar 470, São Paulo 05403-000, Brazil; 4Division of Molecular Immunology, Medical Mycology Research Center, Chiba University, Chiba 263-8522, Japan; 5Hospital das Clínicas of the University of São Paulo (HCFMUSP), University of São Paulo, São Paulo 05403-000, Brazil; 6Center of Immunology, Adolfo Lutz Institute, São Paulo 05403-000, Brazil

**Keywords:** SARS-CoV-2, COVID-19, T-lymphocytes, antiviral response, cytotoxic factors

## Abstract

COVID-19, the infectious disease caused by SARS-CoV-2, has spread on a pandemic scale. The viral infection can evolve asymptomatically or can generate severe symptoms, influenced by the presence of comorbidities. Lymphopenia based on the severity of symptoms in patients affected with COVID-19 is frequent. However, the profiles of CD4+ and CD8+ T cells regarding cytotoxicity and antiviral factor expression have not yet been completely elucidated in acute SARS-CoV-2 infections. The purpose of this study was to evaluate the phenotypic and functional profile of T lymphocytes in patients with moderate and severe/critical COVID-19. During the pandemic period, we analyzed a cohort of 62 confirmed patients with SARS-CoV-2 (22 moderate cases and 40 severe/critical cases). Notwithstanding lymphopenia, we observed an increase in the expression of CD28, a co-stimulator molecule, and activation markers (CD38 and HLA-DR) in T lymphocytes as well as an increase in the frequency of CD4+ T cells, CD8+ T cells, and NK cells that express the immunological checkpoint protein PD-1 in patients with a severe/critical condition compared to healthy controls. Regarding the cytotoxic profile of peripheral blood mononuclear cells, an increase in the response of CD4+ T cells was already observed at the baseline level and scarcely changed upon PMA and Ionomycin stimulation. Meanwhile, CD8+ T lymphocytes decreased the cytotoxic response, evidencing a profile of exhaustion in patients with severe COVID-19. As observed by t-SNE, there were CD4+ T-cytotoxic and CD8+ T with low granzyme production, evidencing their dysfunction in severe/critical conditions. In addition, purified CD8+ T lymphocytes from patients with severe COVID-19 showed increased constitutive expression of differentially expressed genes associated with the caspase pathway, inflammasome, and antiviral factors, and, curiously, had reduced expression of TNF-α. The cytotoxic profile of CD4+ T cells may compensate for the dysfunction/exhaustion of TCD8+ in acute SARS-CoV-2 infection. These findings may provide an understanding of the interplay of cytotoxicity between CD4+ T cells and CD8+ T cells in the severity of acute COVID-19 infection.

## 1. Introduction

COVID-19, a disease caused by SARS-CoV-2 (Severe Acute Respiratory Syndrome Coronavirus 2), has spread on a pandemic scale since the first case was reported in Wuhan, China, in 2019 [1]. According to data from the World Health Organization (WHO), as of 17 June 2022, more than 535 million individuals have been infected worldwide, and 6.3 million have died [2]. Most patients with the disease develop mild or moderate symptoms; however, a portion of patients progress to severe pneumonia and Severe Acute Respiratory Syndrome (SARS), septic shock, and/or multiple organ failure [3,4]. SARS-CoV-2 can infect a wide range of cells, including cardiocytes and endothelial testicular cells, and the bile duct [5]. However, the expansion of vaccination protocols as well as booster doses is contributing to reductions in symptoms, severity of infection, and deaths [6].

An effective immune response against viral infections is mediated by the activation of cytotoxic T cells that can eliminate infected cells. Both CD4 and CD8 T cells have cytotoxic activities contributing to the elimination of virally infected cells, including innate cells such as Natural Killer cells [7]. In this context, in the lung tissue of patients with severe disease, an intense infiltrate of CD4+ and CD8+ T cells is observed, with strong expression of granzyme B [8]. In the context of COVID-19, a pro-inflammatory Th1-cytotoxic response against SARS-CoV-2 spike, membrane, and nucleocapsid proteins is also reported [9]. A controversial role is played in COVID-19 by CD8+ T cells with reduced levels of CD107a, IFN-γ, IL-2, and granzyme B compared to healthy cells [10], or by CD8+ T cells with increased production of granzyme A and B and perforin during COVID-19 [11].

A hallmark of SARS-CoV-2 acute infection is a pronounced reduction in the numbers of CD4+ T lymphocytes, CD8+ T lymphocytes, B lymphocytes, and NK cells, which are associated with higher mortality rate [1,12,13].

Although T-cell responses are important in eliminating viral respiratory infections [14], exacerbation or dysfunctional responses may contribute to the pathogenesis of COVID-19. Increased proinflammatory cytokine levels are of great importance for the recruitment of immune cells to the site of infection and for the fight against the virus, but systemic immune hyperactivation due to SARS-CoV-2 infection can promote loss of negative feedback in the immune system, generating an overproduction of inflammatory cytokines [15]. The proinflammatory environment and constant cellular activation during SARS-CoV-2 infection can also promote exhaustion, generating immune dysfunction with increased PD-1 expression. However, PD-1 expression has also been linked to avoiding exacerbated responses, in addition to suggesting cellular exhaustion in COVID-19 [16,17].

Although the host’s innate immune system possesses elaborate antiviral defense programs, viruses continually develop strategies to evade the host’s immune response. SARS-CoV-2 proteins can antagonize type I IFN response and signaling [18] by mechanisms such as suppression of STAT2 phosphorylation and inhibition of STAT1 nuclear translocation, among others [19]. However, the antagonistic mechanisms of these viral proteins and their contributions to the development and transmission of COVID-19 are poorly understood [20]. COVID-19 patients admitted to the ICU (Intensive Care Unit) have also shown higher levels of CD95 expression on T cells as well as sFasL in plasma, both of which are associated with higher levels of caspase activation; in addition, transcripts of pro-apoptotic members of the Bcl-2 family, Bax and Bak, are upregulated. This indicates that CD4 and CD8 T cells from COVID-19 patients are more likely to die from apoptosis [21].

This study aimed to phenotypically and functionally evaluate T lymphocytes in moderate and severe/critical cases of COVID-19. Data indicate enhancement of CD4+ and CD8+ T-cytotoxic profiles in severe COVID-19 patients, whereas CD4+ T cells are less activated than CD8+ T cells. Dysfunctional cytotoxicity of CD8 T cells has been linked with the expression of genes associated with the caspase pathway as well as inflammasome, showing them to be more prone to death.

## 2. Materials and Methods

### 2.1. Casuistic

Blood samples in EDTA and heparin from a ward and the ICU of the Central Laboratory Division (DLC) of the Hospital das Clínicas da Faculdade de Medicina da Universidade de São Paulo (HCFMUSP) were used. The study was approved by the ethics committee (CAAE 30800520.7.0000.0068-2020). Complete blood samples (in EDTA) were kept at 4 °C and were used the next day. Heparin samples were used on the day of collection. As an inclusion criterion, it was necessary to confirm the diagnosis of COVID-19 through the detection of SARS-CoV-2 RNA by reverse transcription polymerase chain reaction (RT-PCR). Patients over the age of 75 years and those who did not test positive for SARS-CoV-2 were excluded from the study. In addition, 25 healthy individuals negative for SARS-CoV-2 by RT-PCR were included as a control.

The cohort of 62 patients infected with COVID-19 included 34 males and 28 females. Patients were categorized based on the WHO classification (WHO, 2020): Hospitalized patients without oxygen therapy or receiving oxygen by mask or nasal cannula were considered “moderate.” Patients admitted under non-invasive ventilation or high-flow oxygen were considered “severe,” and patients admitted under invasive ventilation without or with support from another organ (e.g., extracorporeal membrane oxygenation (ECMO) or replacement therapy) were considered “critical” cases. Severe and critical cases were evaluated together, and critical cases were flagged. The EDTA samples were obtained from May to July 2020, and the heparin samples were obtained between May and July 2021.

### 2.2. Phenotypic Analysis of CD4+ and CD8+ T Lymphocytes in Peripheral Blood by Flow Cytometry

For phenotypic characterization, 100 μL of whole blood collected in EDTA tubes was incubated with a viability marker LIVE/DEAD Fixable Red Dead Cell Stain Kit (Invitrogen, Carlsbad, CA, USA) for 20 min and subsequently incubated with surface antibodies (CD3-BV506 clone SK7, CD4-Pe-Cy7 clone SK2, CD8-APC-Cy7 clone MEM-31, CD28-FITC clone CD28.2, CD38-PerCP-Cy5.5 clone HIT2, and HLA-DR-V500 clone G46-8) for 30 min at room temperature. After this period, the samples were washed and fixed with 4% formalin for 15 min, and red blood cells were lysed with FACS Lysing Reagent (BD Biosciences, Franklin Lakes, NJ, USA) for 15 min at room temperature. Then, the cells were washed and resuspended in phosphate buffer (PBS). To evaluate PD-1 expression in CD4+ and TCD8+ T lymphocytes, cell staining was performed from PBMCs incubated with surface antibodies (CD3-BV605 clone SK7, CD4-V500 clone RPA-T4, CD8-V450 clone RPA-T8, CD56-AL-100 clone B159, and PD-1-APC clone MIH4). Approximately 100,000 events were acquired per sample on a Fortessa LSR flow cytometer (BD Biosciences). Fluorescence Minus One (FMO) labeling was used, which labels the sample with all the Antibodies (Ab) except the Ab to be analyzed. Data analysis was performed using FlowJo™ software V10. The analysis strategies for T lymphocytes and PD-1 expression are illustrated in Appendix A.

### 2.3. Cytotoxic CD4+ and CD8+ T-Cell Profile

The volume of blood in a heparinized tube was diluted in saline solution and centrifuged in Ficoll–Hypaque solution (Amersham Pharmacia Biotech, NJ, USA) for 20 min at 2200 rpm. Afterward, the PBMCs obtained were washed twice in saline solution for 10 min at 1200 rpm.

To assess the cytotoxic profile of CD4+ and CD8+ T lymphocytes, 1 × 10^6^ PBMCs were distributed in a 48-well microplate (Costar, Cambridge, MA, USA) in RPMI culture medium (Gibco, Carlsbad, CA, USA) containing 5% AB human serum (Sigma-Aldrich, St. Louis, MO, USA). Cells were stimulated with 30 ng/mL PMA and 500 ng/mL Ionomycin (Sigma-Aldrich) in the presence of CD107a PE-Cy 5 antibody (1:2500) (BD Biosciences) and 10 µg/mL of Brefeldin A (Sigma-Aldrich) and were incubated at 37 °C at 5% CO_2_ for six hours. Subsequently, cells were collected, washed, and resuspended in PBS with the viability marker LIVE/DEAD Fixable Red Dead Cell Stain Kit (Invitrogen) and incubated for 20 min at room temperature. Then, the cells were washed with PBS and incubated with antibodies for 30 min in the dark at room temperature (CD3-BV605 clone SK7, CD4-V500 clone RPA-T4, CD8-APC-Cy7 clone MEM-31, CD56-AL-700 clone B159, Granzyme B-V450 clone GB11, CD107a-Pe-Cy5 clone H4A3, TNF-Pe-Cy7 clone Mab11, Perforin-PE clone 27-35, and IFN-y-FITC clone B27). Subsequently, the cells were fixed for 15 min with 4% formalin and incubated for 30 min with the antibodies for intracellular labeling together with saponin. After this period, the cells were washed and resuspended in PBS and, after 18 h, were acquired in a Fortessa LSR flow cytometer (BD Biosciences). Approximately 100,000 events were sampled. Data analysis was performed using the FlowJo™ software. The gating strategy used is illustrated in Appendix A.

### 2.4. PCR Array of the Expression of Antiviral Factors in Purified CD8+ T Lymphocytes

CD8+ T lymphocytes were isolated from PBMCs using the EasySep™ Human CD8 Positive Selection Kit II on a STEMCELL EasyEights magnet following the manufacturer’s instructions. After isolation, the CD8+ T lymphocytes obtained were quantified, and the degree of purity was above 80% (CD3-BV605 clone SK7 and CD8-V450 clone RPA-T8). An RNeasy Plus Mini Kit (Qiagen, Valencia, CA, USA) was used to extract total RNA from the samples following the manufacturer’s recommendations. RNA levels were measured using the NanoDrop ND-1000 spectrophotometer (Thermo Scientific, MA, USA). For cDNA synthesis, an RT2 First Strand kit (Qiagen) was used.

For real-time PCR reactions, an RT2 SYBR Green/ROX qPCR Master Mix (Qiagen) was used, which contains SYBR Green as a fluorophore and ROX as a passive reference. The PCR array kit used was PAHS122Z Antiviral Response (Qiagen) in accordance with the manufacturer’s instructions. Gene expression data from purified CD8+ T lymphocytes were analyzed using the comparative Ct method. For normalization of the data, an average value of the following reference genes was used: ACTβ (beta-actin), GAPDH (glyceraldehyde 3 phosphate dehydrogenase), HPRT1 (hypoxanthine phosphoribosyl transferase 1), and RPLP0 (ribosomal protein, large, P0).

### 2.5. Statistical Analysis

To perform statistical analysis and graphical representation of the data, Graph Pad Prism 9 (Graph Pad Software Inc., La Jolla, CA, USA) was used. Results were expressed as the median and interquartile range (IQR). Analysis of variance was performed using the one-way ANOVA test with the non-parametric Kruskal–Wallis test to compare the three groups of data. For comparative analyses between two groups, the Mann–Whitney test was used, and for correlation analysis, the Pearson test was used. A level of significance was considered when *p* ≤ 0.05.

## 3. Results

There was decreased frequency of activated CD4+ and CD8+ T lymphocytes and increased PD1 expression in patients with SARS-CoV-2 infection.

In the cohort, there were 62 patients infected with COVID-19 (34 males and 28 females) with diagnoses confirmed by PCR; all of whom were included in this study. We also included 25 uninfected patients. The demographic data of the individuals participating in the study are summarized in Figure 1. We observed that 90.9% of patients with moderate infection were discharged, and 9.09% died. For patients with severe/critical infection, 62.5% were discharged, 32.5% died, 2.5% were transferred to other institutes, and we were unable to access the medical records for 2.5%. In individuals with both moderate and severe/critical infections, the occurrence of systemic arterial hypertension (SAH) and/or cardiovascular disorders was prevalent (Figure 1).

Figure 2A–C show the percentage of total lymphocytes and CD4+ and CD8+ T cells, respectively, out of the total living cells. There was a reduction in the frequency of total lymphocytes in patients with severe/critical disease in relation to individuals without the infection and patients with moderate disease (Figure 2A). As for CD4+ and CD8+ T lymphocytes, a percentage reduction was observed in infected patients compared to non-infected ones (Figure 2B,C). In the CBC assessment, the neutrophil/lymphocyte ratio (Figure 2D) was increased in patients with severe disease compared to uninfected patients with moderate disease, indicating leukocytosis (Figure 2E) from neutrophilia and lymphopenia in severe cases of infection. Figure 2F–H show the total number of lymphocytes, T CD4+, and T CD8+ in mm^3^, respectively, reflecting, in fact, lymphopenia in lymphocyte subpopulations.

Subsequently, we evaluated the frequency of total lymphocytes and CD4+ and CD8+ T cells in patients with moderate and severe/critical disease separated by SARS-CoV-2-positive PCR based on the days (from 1–7 days and from 8–20 days PCR positive). As seen in Figure 2J,K, an increase in the frequency of CD4+ and CD8+ T lymphocytes, respectively, out of the total living cells was detected in patients with moderate disease after the seventh day of positive PCRs in relation to patients with positive PCRs before this period. These data indicated a recovery in the frequency of CD4+ and CD8+ T lymphocytes after the seventh day of positive PCRs in moderate patients that was not observed in individuals with severe/critical infection, indicating the persistence of lymphopenia with the severity of the infection. In Appendix A, the numbers of T CD4+ and T CD8+ lymphocytes in different periods of being positive PCR are shown; however, there was no difference, as this recovery was observed only in the evaluation of the frequency.

After evaluating the frequency of CD4+ and CD8+ T lymphocytes, we analyzed the expression of activation markers in these cell populations of patients infected with SARS-CoV-2. Figure 3A,D show the expression of the CD38+ marker on CD4+ and CD8+ T lymphocytes, respectively. HLA-DR is a cell surface glycoprotein encoded by the HLA-DR region of the major histocompatibility complex expressed at high levels in APCs. However, HLA-DR expression in effector T lymphocytes is described in some viral infections and autoimmune diseases as a marker of activation. We observed a reduction in the percentage of CD4+ CD38+ T lymphocytes (Figure 3A).

In contrast, an increase in CD38+ expression in CD8+ T lymphocytes was observed in patients compared to uninfected individuals (Figure 3D). The increase in CD38+ in CD8 T lymphocytes was also observed in the evaluation of MFI (Appendix A). We observed an increase in the percentage of CD4+ T lymphocytes (Figure 3B) and CD8+ T lymphocytes (Figure 3E) that express HLA-DR in COVID-19 patients, regardless of the severity of symptoms, compared to uninfected individuals. CD38 is a glycoprotein with ectoenzymatic functions and is expressed on mature T-lymphocyte subtypes. These cells have an activated phenotype associated with reduced proliferative capacity. However, they have the ability to produce IL-2 and IFN-γ.

In Figure 3C,F, the expression of CD28+ on CD4+ and CD8+ T lymphocytes is represented. We observed an increase in the frequency of CD4+ CD28+ T lymphocytes in patients with severe/critical disease compared to their controls (Figure 3C).

To assess the exhaustion phenotype in critically ill patients with COVID-19, we verified an increase in the frequency of CD4+ and CD8+ T cells that express PD-1 in severe/critical patients compared to uninfected individuals. In the MFI analysis, we did not observe any change in the expression of PD-1 (Appendix A). Importantly, increased PD-1 expression does not necessarily indicate exhaustion. It is necessary to evaluate other markers in addition to PD-1, including indicators of cellular activation, persistent infection, and continuous stimulation of T cells.

Taken together, these results indicate that lymphopenia worsens with the severity of symptoms. CD4+ T cells are less activated, while CD8+ T cells are more activated, and lymphocytes show an exhaustion phenotype in COVID-19, which may contribute to the pathogenesis of the infection.

### 3.1. Dysfunctional TCD8+ Lymphocytes and Generation of Cytotoxic TCD4+ Lymphocytes in COVID-19

The functionality of T cells and NK cells, as well as the production of granzyme, perforin, CD107a, IFN-γ, and TNF in PBMCs from uninfected individuals and patients with severe/critical COVID-19, was evaluated by flow cytometry at baseline or after stimulation with PMA and Ionomycin.

It is noteworthy that in the basal condition (not stimulated), higher production of granzyme, IFN-γ, and CD107a was observed in CD4+ T lymphocytes of patients with severe infection compared to uninfected individuals (Figure 4A and Appendix A). After stimulation, no difference was observed among the groups (Figure 4B and Appendix A). The data show that CD4+ T cells already had an ex vivo alteration and were little affected by the stimulus. The results indicate that SARS-CoV-2 infection can induce the generation of cytotoxic CD4+ T cells with the presence of IFN-γ, granzyme, and CD107a.

As for CD8+ T lymphocytes, at basal condition there was a decrease in perforin production but an increase in IFN-γ and TNF in severe/critical cases compared to uninfected individuals (Figure 4C). When cells were stimulated with PMA and Ionomycin, increased production of IFN-γ was observed, with a drop in production of perforin and CD107a in critically ill patients (Figure 4D). The data emphasize that CD8+ T cells, despite the expression of IFN-γ, show an altered expression of degranulation and cytotoxic markers in COVID-19.

To better understand the ability of lymphocyte subtypes to produce cytotoxic factors in severe/critical COVID-19 infection, we performed a dimensionality reduction assessment, allowing us to explore populations by the t-SNE technique. To perform t-SNE, singlets, single cells, and live cells were selected, and the lymphocyte gate was subsequently performed. A total of 1000 events were selected to analyze population clustering. In the t-SNE evaluation, a total of seven clusters were identified that shared common characteristics (Figure 5A,B). With the t-SNE data, we classified the populations into four groups: One (basal uninfected), Two (basal severe/critical COVID-19), Three (stimulated uninfected), and Four (stimulated severe/critical COVID-19). Figure 5D shows the overlap of these populations. Figure 5E shows the clusters corresponding to the basal situation of the uninfected patients (red) and the patients with severe COVID-19 (blue), while Figure 5F shows the clusters corresponding to the stimulated cells of the uninfected patients and the patients with severe COVID-19.

We observed that in both basal and stimulated situations, there was a lower intensity of Cluster 6 (CD8 and granzyme expression more prominent) in patients with severe symptoms (Figure 5C and Appendix A). These data relate to the reduction in the production of cytotoxic factors in patients with severe/critical conditions of COVID-19 observed in conventional cytometry, evidencing dysfunction of CD8+ T lymphocytes.

Taken together, these results demonstrate that in COVID-19 there is an induction of the CD4+ T-lymphocyte cytotoxic response in severe/critical cases. Interestingly, the production of perforin and CD107a was altered in CD8+ T lymphocytes after stimulation, showing cytotoxic dysfunction. On the other hand, despite the increase in the baseline condition, CD4+ T lymphocytes were responsive to stimulation via Protein Kinase C (PKC) and were balanced at the levels of the uninfected group.

### 3.2. Increased Differential Gene Expression Associated with the Caspase and Inflammasome Pathway in TCD8 Lymphocytes from Patients with Severe COVID-19

As observed in the previous results, CD8+ T lymphocytes from individuals affected with severe COVID-19 were found to have increased PD-1 expression as well as reduced production of cytotoxic factors. There was also an increase in the production of IFN-γ and TNF, indicating an exhausted and inflammatory profile. To evaluate whether this observed inflammatory and exhausted profile affected the expression of antiviral factors and the expression of signaling molecules, an array PCR of 84 genes associated with these factors was performed.

Figure 6 illustrates a heatmap of the relative gene expression of antiviral factors and constitutive CD8+ T-lymphocyte signaling pathways from five patients with severe COVID-19 and from four uninfected patients. Overall, we observed greater intensity of gene expression in COVID-19-positive patients compared to controls (Figure 6B). Figure 6A,B show the Volcano plot and column graph of these data, respectively, highlighting the up- and downregulated genes in severe SARS-CoV-2 infection relative to uninfected individuals. We observed that *CASP8, CASP10, PYCARD, CARD9, IL-18, CD80, TLR9, IRF3, IRAK1, IKBKB, OAS2,* and *MX*1 genes were upregulated and *TNF* was downregulated. The expression of genes CASP8, CASP10 was related to the caspase and apoptosis pathway, while PYCARD, CARD9, and IL-18 were related to inflammatory responses to infection. Thus, increased expression of these genes was associated with an increase in inflammatory markers observed in the phenotypic analysis, as well as a reduction in the number of lymphocytes evidenced in SARS-CoV-2 infection. Overall, severe COVID-19 was shown to induce genes associated with caspase pathways and apoptotic processes, especially involving the extrinsic pathway and inflammasome-associated factors. Severe COVID-19 was also shown to upregulate the *CD80* gene (also called B7.1) (Figure 7G) and *TLR9* (Figure 7H). It upregulated genes such as *IRAK1, IKBKB,* and *IRF3* associated with cell signaling pathways after pathogen recognition (Figure 7I,J,L). We also evidenced increased expression of viral *OAS* (Figure 6E) and *MX1* (Figure 6F) in patients with severe COVID-19. Finally, we demonstrated that *TNF* was downregulated in TCD8 lymphocytes in severe SARS-CoV-2 infection (Figure 7M).

Overall, the data show increased expression of genes associated with signaling in response to viral stimulus and inflammatory and apoptotic pathways in severe COVID-19 as being related to the pathogenesis of the SARS-CoV-2 infection.

## 4. Discussion

There are gaps in the knowledge about the pathogenesis of acute SARS-CoV-2 infection. T cells play an important role in the elimination of viruses and for long-term protection against infections; however, they may exhibit a dysfunctional profile and/or collaborate with tissue damage in target organs. In this context, functional assessments of CD4+ and CD8+ T cells, as well as their implications, become relevant in SARS-CoV-2 infection.

In the cohort of patients infected with SARS-CoV-2 at Hospital das Clínicas–FMUSP, we classified patients with moderate and severe/critical symptoms. We observed that 90.9% of patients with moderate symptoms were discharged, and 9.09% died. In severe/critical cases, 62.5% were discharged, and 32.5% died. Patients with severe manifestations such as pneumonia, hypoxemia, SARS, sepsis and septic shock, cardiomyopathy, arrhythmia, and acute kidney injury required hospitalization and supportive care and were more likely to die as a result of the disease [22]. The most frequent comorbidities observed in our cohort were, sequentially, systemic arterial hypertension and cardiovascular disorders, obesity, and diabetes mellitus, and they occurred similarly between moderate and severe/critical clinical conditions. In chronic comorbidities, prolonged pro-inflammatory state and dysfunction of innate and adaptive immunity were the drivers of worse clinical outcomes in patients infected with SARS-CoV-2 [23]. In patients with obesity and diabetes, ACE2 expression was found to be upregulated, thus increasing susceptibility to SARS-CoV-2 infection [24]. These data may help us to understand the relationship between comorbidities and the severity of SARS-CoV-2 infection.

Leukocytosis was more pronounced in patients with a severe/critical condition, which was associated with neutrophilia and an increase in the neutrophil–lymphocyte ratio (NLR), which highlighted lymphopenia in these patients. The NLR has been indicated as a predictor of the severity of COVID-19 since the infiltration of neutrophils in the lung and neutrophilia correlate with the histopathological lesion [25]. After the first week of infection, we observed an increase in the frequency of T lymphocytes in patients with moderate infection, but we did not find this in severe/critical cases. Patients with prolonged hospitalization due to COVID-19 did not show recovery of B-lymphocyte and CD4 T-lymphocyte counts [26]. These data indicate a slower restoration capacity of lymphocyte frequency in the most severe cases of infection.

We observed a reduction in the frequency of total lymphocytes, T-CD4+ and T-CD8+, more markedly in patients with severe/critical conditions. In fact, reduced lymphocyte frequency is a recurrent feature in SARS-CoV-2 infection, with reduced numbers of CD4+ T lymphocytes, CD8+ T lymphocytes, B lymphocytes, and NK cells found to have a strong association with the mortality rate and gravity [1,12]. Several mechanisms may be associated with the occurrence of lymphopenia in SARS-CoV-2 infection, such as attraction of T and NK cells to sites of infection and sequestration of lymphocytes in target organs [27,28,29]; SARS-CoV-2 infection in human T cells [30]; and higher expression of p53 in PBMCs of patients with COVID-19, which leads to apoptosis [31]; among others. In this context, higher levels of CD95 expression on T cells and sFasL in plasma, both associated with higher levels of caspase activation, have been described in patients admitted to the ICU because of COVID-19. Genes such as Bax and Bak are upregulated, indicating that CD4 and CD8 T cells from COVID-19 patients are more likely to die from apoptosis [21].

In the evaluation of CD4+ T lymphocytes, we observed an increase in the frequency of HLA-DR and CD28 expression and a reduction in CD38+ expression. It seems that CD4+ T-cells in severe COVID-19 patients were in an activated status but not chronically, whereas they showed upregulation of PD1 expression. In addition to suggesting cellular exhaustion, PD-1 expression has been linked to avoiding exacerbated responses [16,17]. Thus, the increase in PD-1 expression did not necessarily indicate cellular exhaustion, but rather a way to avoid intense immune responses at the beginning of cellular activation, thus being a marker of activation. In addition, the expression of PD-1 alone was not sufficient to define cell exhaustion, requiring the evaluation of other markers for confirmation. When they were not stimulated, we detected an increase in the frequency of granzyme, IFN-γ, and CD107a of CD4+ T lymphocytes in patients with severe/critical COVID-19. In t-SNE, we also observed qualitatively greater intensity of the cluster that characterizes CD4+ T lymphocytes with a cytotoxic profile. Our data show cytotoxic factors such as CD107a, granzyme, and IFN-γ, representative of the T-CD4+ CTL function, which are generated in COVID-19, as already described for other viral infections [7]. In the context of COVID-19, a Th1-cytotoxic pro-inflammatory response against the spike, membrane, and nucleocapsid proteins of SARS-CoV-2 has also been reported [9]. However, the differences between patients and unexposed individuals were mainly quantitative rather than qualitative, suggesting that the cells identified were not unique to COVID-19 but may have represented a common cellular phenotype of antiviral T cells [32]. On the other hand, a reduction in the production of IFN-γ and granzyme A in CD4+ T lymphocytes in patients with COVID-19 was also reported in another study [33], indicating a functional change.

In our cohort of patients, CD8+ T cells showed increased expression of CD38, HLA-DR+, and CD28+. Moreover, the increase in IFN-γ in basal and stimulated levels of CD8+ T lymphocytes in patients with severe/critical condition in relation to controls stood out, with a decrease in CD107a and perforin and increased expression of PD1. This finding was also verified in the t-SNE evaluation, with the reduced cluster of cells expressing a greater intensity of CD8 and granzyme. This indicates that CD8+ T cells are dysfunctional in severe/critical conditions of infection. Decreased production of CD107a+, IFN-γ+, IL-2+, and granzyme B+ was also described in CD8+ T lymphocytes in COVID-19 [10]. In contrast, SARS-CoV-2 infection was found to induce a cytotoxic response of CD8+ T cells characterized by the simultaneous production of granzyme A and B and perforin [11].

Notwithstanding the connection between IFN-γ production by CD8+ T lymphocytes and disease severity, antiviral factor and apoptotic molecule expressions remain unknown. We found equilibrated expression of ISGs such as *ISG15, IRF7, STAT1, IFNA1, IFNA2,* and *IFNAR1* by PCR array in CD8+ T lymphocytes. We also evidenced an increase in the expression of *OAS2* and *Mx1*, which play an important role in defense against viral infections by catalyzing the synthesis of 2′-5′-oligoadenylate for the activation of RNase L [34] or inhibiting the infection by blocking viral transcription and replication, respectively [35,36]. It has been reported in SARS-CoV-2 infection that the virus can suppress the type I IFN response by evasion mechanisms such as ubiquitination of cytosolic sensors and inhibition of translocation of nuclear factors by decreasing *STAT1* phosphorylation, among other mechanisms that are still not fully understood [19,37,38]. Despite the presence of antiviral expression by CD8 T cells, in severe cases of COVID-19 that were induced by IFN-g, the dysregulation in cytotoxic component contents disabled them from effective cytotoxic action.

It was noticeable that the increased DEGS expression was associated with caspase pathways and apoptotic processes such as *CASP8, CASP10, PYCARD, CARD9,* and *IL-18* genes. These components evidenced a pro-apoptotic process, which, in turn, may contribute to the lymphopenia. It has been described that, besides CD4+ T cells, CD8 T cells also die in severely-affected COVID-19 patients compared to uninfected individuals [21]. Other studies have also shown increased expression of genes associated with inflammation in macrophages challenged with SARS-CoV-2 [39] and associated with the caspase pathway and apoptosis [40]. The inflammasome (NLRP3, associated with IL-18 activation) was reported to be activated during SARS-CoV-2 infection and has been proposed as an indicator of COVID-19 disease severity, predicting the release of pro-inflammatory cytokines that lead to dysregulated immune responses and tissue damage [41].

Increased expression of TLR9 and CD80 was also observed in patients with severe/critical disease. TLR9 recognizes motifs rich in unmethylated cytosine–phosphate–guanine (CpG) sequences and has been described in vulnerable patients; TLR9 activation may be a silent driving force but helps explain the SARS-CoV-2 aggravation of hyperinflammation [42]. Although B7 in APCs has a well-recognized role in T cell co-stimulation, B7 expression in lymphocytes was also described as conferring new functional properties on T cells, such as prolonged lifespan and the ability to provide co-stimulatory signals with autologous T lymphocytes [43].

Interestingly, we showed that TNF was downregulated in CD8+ T lymphocytes in severe SARS-CoV-2 infection. TNF is associated with the effector function of cytotoxic cells such as NK and CD8+ T lymphocytes, but it also displays an immunosuppressive role, facilitating the biological activity of Tregs and myeloid-derived suppressor cells [44]. The presence of TNF in CD8+ T cells had a negative effect, inducing the death of these cells; thus, the reduction in gene expression may indicate an attempt by CD8+ T lymphocytes to maintain their effector function and prevent cell death. In addition, CD8+ T cells with an exhausted profile may modulate cellular apoptosis.

## 5. Conclusions

Taken together, these findings highlight the involvement of T cells in the immunopathogenesis of COVID-19. T lymphocytes showed an activated and exhausted phenotype, according to the severity of symptoms. In the cytotoxic profile, an increase in the response of CD4 cells has been evidenced already in ex vivo condition. CD8+ T lymphocytes showed a more dysfunctional and exhausted profile in the cytotoxic response, with the induction of antiviral gene expression. These findings may provide a better understanding of factors associated with infection severity. More studies are needed to assess the involvement of these cells in the course of COVID-19 disease.

## Figures and Tables

**Figure 1 cells-11-03359-f001:**
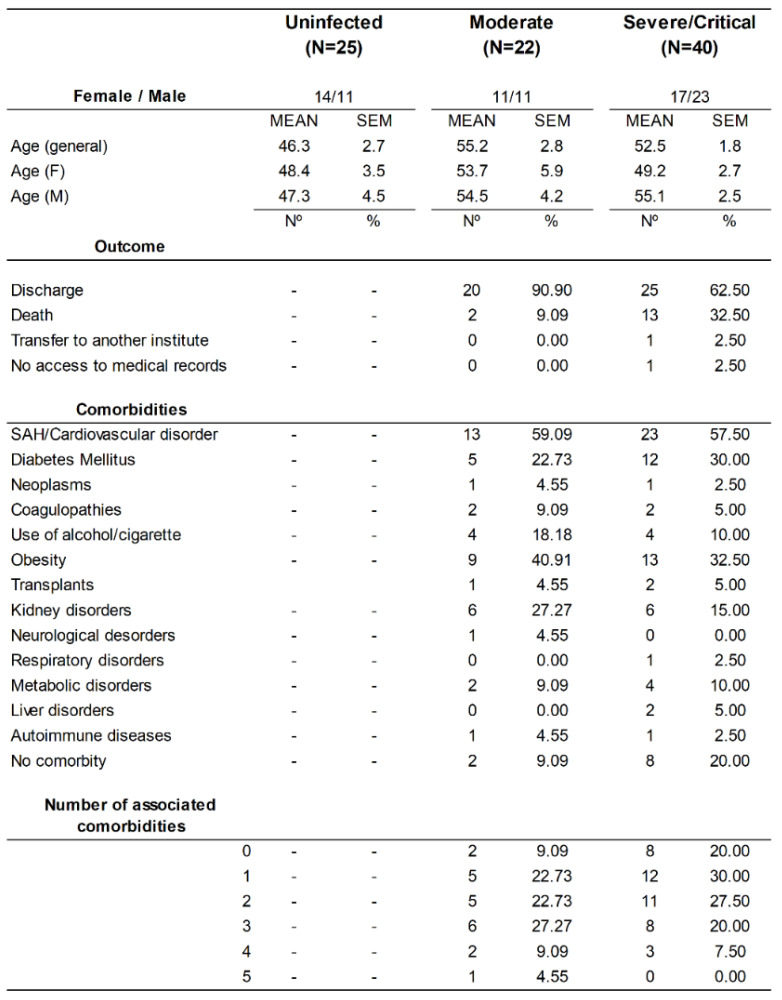
Demographic data of patients affected by SARS-CoV-2 and uninfected individuals. Data on age, outcome, and comorbidities of uninfected individuals and patients with COVID-19 (moderate and severe/critical): M—male; F—female; N—sample number; and SAH—systemic arterial hypertension.

**Figure 2 cells-11-03359-f002:**
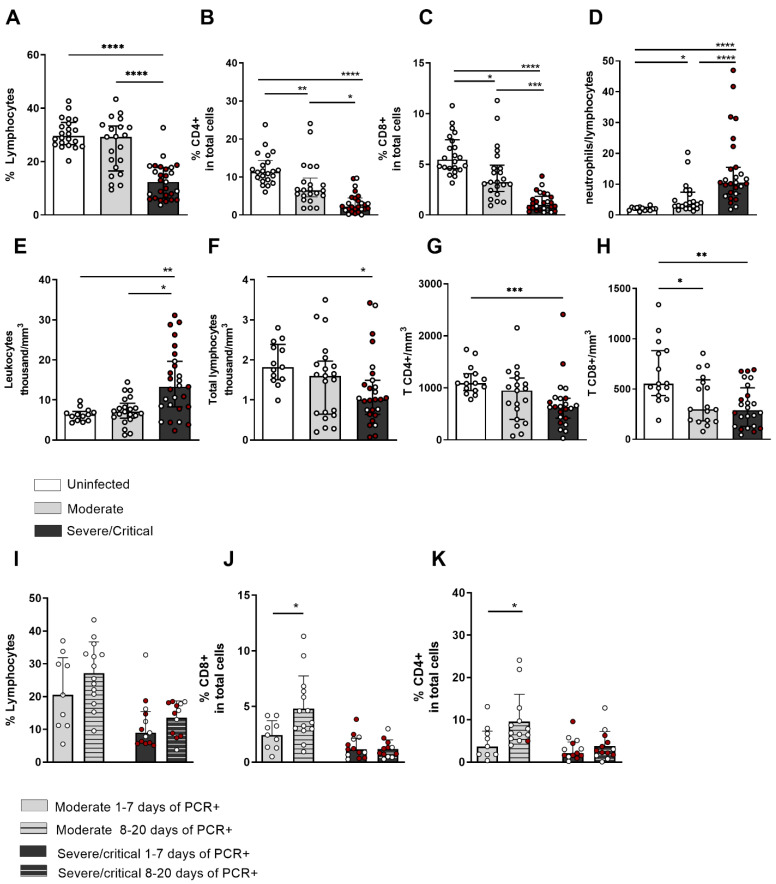
Decreased numbers of CD4+ and CD8+ T lymphocytes in patients with severe/critical disease from SARS-CoV-2 infection. The graphs show the frequency of cells within living cells: (**A**) total T lymphocytes, (**B**) CD4+ T lymphocytes, (**C**) CD8+ T lymphocytes, (**D**) neutrophil/lymphocyte ratio (NLR), (**E**) leukocyte number (thousand/mm^3^), (**F**) total lymphocyte number (thousand/mm^3^), (**G**) total number of CD4+ T lymphocytes per mm^3^, (**H**) total number of CD8+ T lymphocytes per mm^3^, (**I**–**K**) frequency of CD4 and TCD8 T lymphocytes from SARS-CoV-2-infected patients with moderate or severe critical disease between 1–7 days positive and 8–20 days positive PCR. Red dots represent patients with critical infection. The bars represent the median and interquartile range. * *p* < 0.05, ** *p* < 0.01, *** *p* < 0.001, and **** *p* < 0.0001.

**Figure 3 cells-11-03359-f003:**
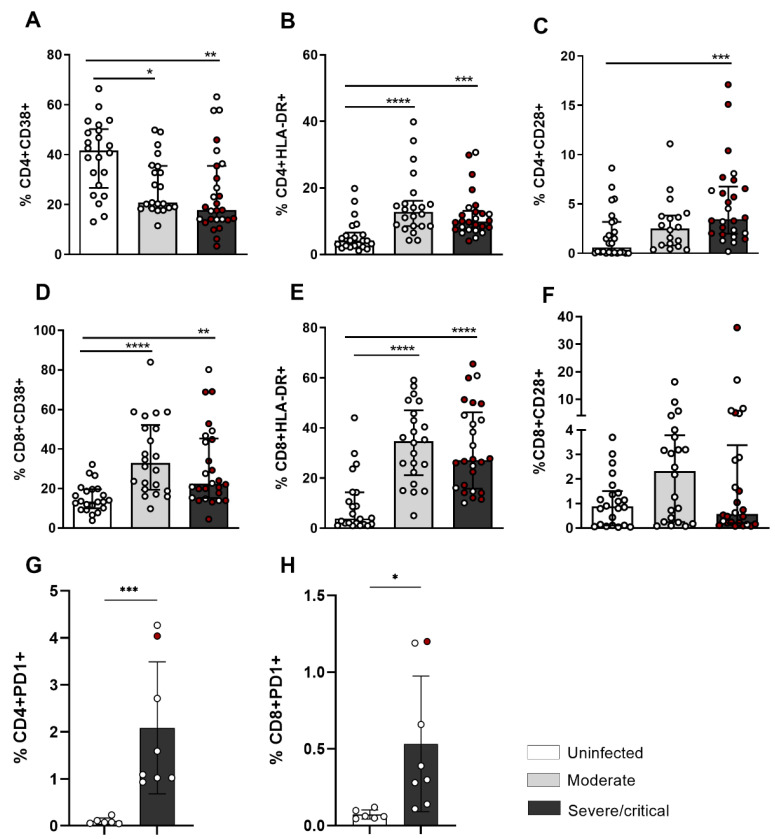
Increased frequency of activation markers and PD-1 expression of CD4+ and CD8+ T lymphocytes of patients infected with SARS-CoV-2. Frequency in CD4+ T lymphocytes: (**A**) CD38+, (**B**) HLA-DR+, and (**C**) CD28+; frequency in CD8+ T lymphocytes of (**D**) CD38+, (**E**) HLA-DR+, and (**F**) CD28+; (**G**) frequency of PD-1 expression within total living cells of CD4+ T lymphocytes; (**H**) frequency of PD-1 expression of CD8+ T lymphocytes. The red dots represent patients with the critical infection. The bars represent the median and interquartile range. * *p* < 0.05, ** *p* < 0.01, *** *p* < 0.001, and **** *p* < 0.0001.

**Figure 4 cells-11-03359-f004:**
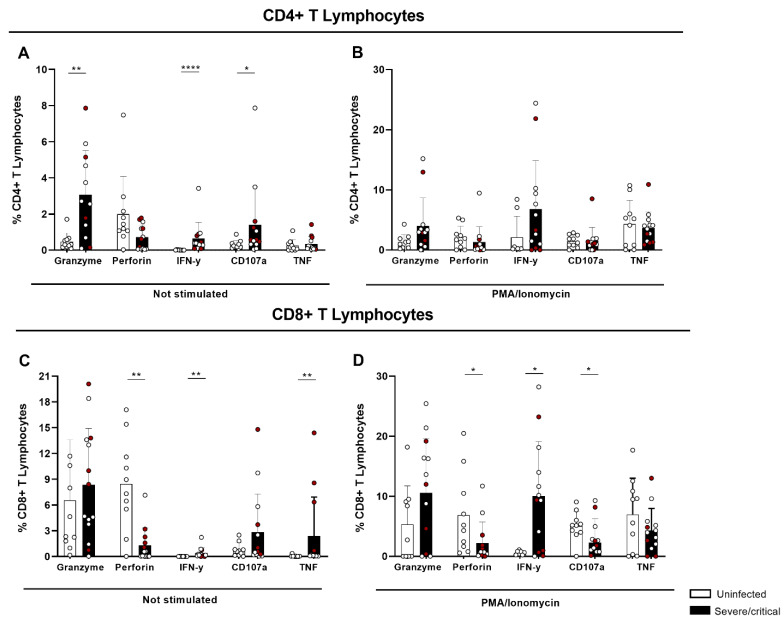
Cytotoxic profile of CD4+ and CD8+ T lymphocytes in SARS-CoV-2 infection. The graphs represent the cytotoxic profile (granzyme A, perforin, CD107a, IFN-γ, and TNF) of CD4+ and CD8+ T lymphocytes from PBMCs from uninfected controls and from patients with severe/critical COVID-19: basal levels of the cytotoxic profile of (**A**) CD4+ T lymphocytes and (**D**) CD8+ T lymphocytes, and upon stimulation with PMA and Ionomycin of (**B**) CD4+ T lymphocytes and (**C**) CD8+ lymphocytes. The red dots represent patients with critical infections. The bars represent the median and interquartile range. * *p* < 0.05, ** *p* < 0.01 and **** *p* < 0.0001.

**Figure 5 cells-11-03359-f005:**
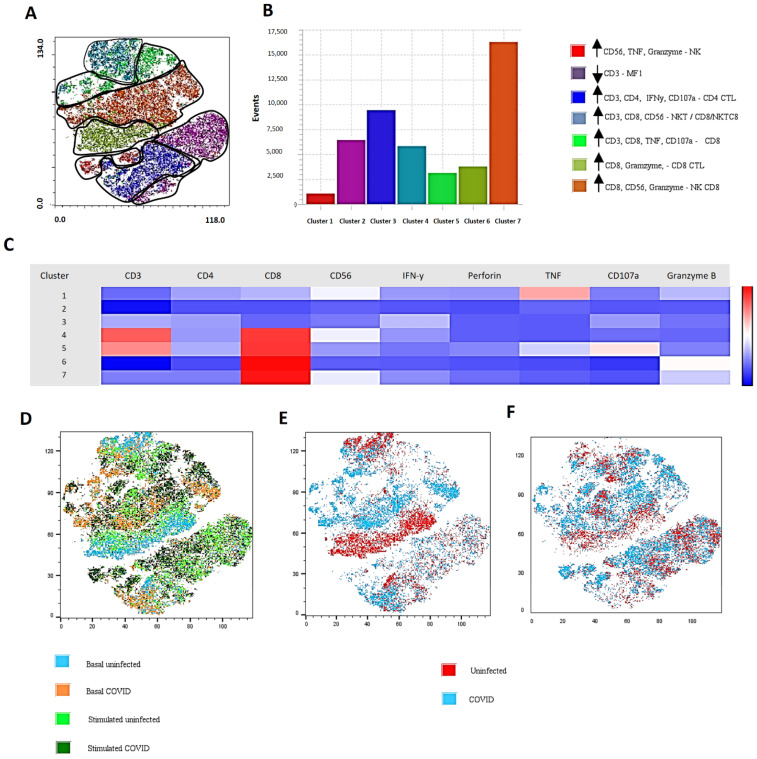
A t-SNE-guided population definition of lymphocyte subtypes from uninfected individuals affected by severe/critical SARS-CoV-2: (**A**) Merged PBMC samples creating a single t-SNE map with the signal strength of lymphocyte phenotypic markers and cytotoxic factors. (**B**) Column graphic representation of the seven clusters identified in the analysis and the corresponding value of events for each of them. (**C**) Heatmap of the MFI of the markers used, identified in each of the clusters. (**D**) Overlay of baseline cell populations from (1) uninfected individuals, (2) basal severe/critical COVID-19, (3) stimulated uninfected, and (4) stimulated severe/critical COVID-19. (**E**) Overlapping cell populations from (1 red) uninfected or (2 blue) infected patients at baseline. (**F**) Overlapping cell populations from (3 red) uninfected or (4 blue) or infected patients in a stimulated situation.

**Figure 6 cells-11-03359-f006:**
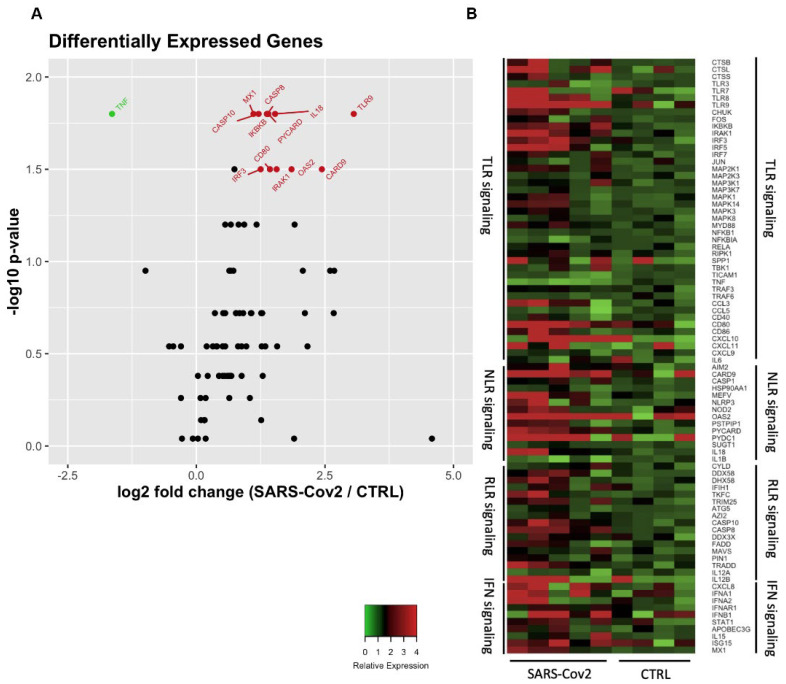
Differentially expressed genes of the signaling pathway of innate immunity and anti−viral factors in CD8+ T lymphocytes in severe COVID-19. (**A**) Expression of antiviral factors by PCR array of 84 genes of CD8+ T lymphocytes from individuals with severe COVID-19 compared to uninfected. The red dots represent the upregulated genes in severe SARS-CoV-2 infection, the green dot represents the downregulated gene, and black dots represent genes that do not change. (**B**) Heatmap of expression of antiviral factors of CD8+ T lymphocytes from uninfected (*n* = 6) and infected individuals with severe COVID-19. Genes are divided into groups according to the associated signaling pathway: Toll-like receptor (TLR), NOD −like receptor (NLR), and RIG−like receptor (RLR).

**Figure 7 cells-11-03359-f007:**
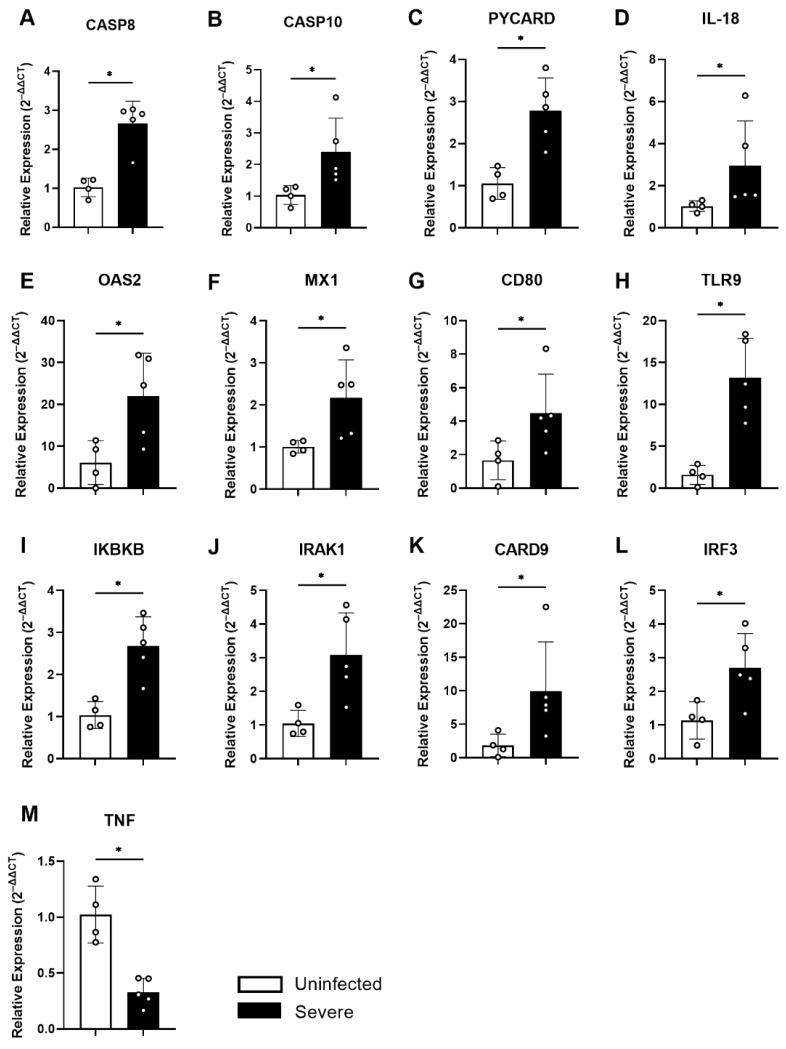
Differentially expressed genes in CD8+ T lymphocytes of severe COVID−19 patients. Differential expression of antiviral factors by array PCR of CD8+ T lymphocyte genes from individuals with severe COVID−19 compared to uninfected individuals. (**A**) Expression of *CASP 8*, (**B**) *CASP10*, (**C**) *PYCARD*, (**D**) *IL-18*, (**E**) *OAS2*, (**F**) *MX1*, (**G**) *CD80*, (**H**) *TLR9*, (**I**) *IKBKB*, (**J**) *IRAK1*, (**K**) *CARD9*, (**L**) *IRF3*, (**M**) *TNF*. The bars represent the median and interquartile range. * *p* < 0.05.

## Data Availability

Not applicable.

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
