# Peer review of "Generation of Cytotoxic T Cells and Dysfunctional CD8 T Cells in Severe COVID-19 Patients"

_cells, 2022, doi:10.3390/cells11213359_

Round 1

Reviewer 1 Report

The aim of this paper is to investigate more deeply into reports of lymphopenia in patients who have contracted SARS-CoV-2 (COVID-19). The Authors take a cohort of patients and stratify them into moderate or severe cases as well as any co-morbidities that may be expressed. Next, the Authors analyse patient blood for lymphocytes and CD4/CD8 T cell numbers.  Further characterisation of CD4 and CD8 T cell populations are shown along with some functional outputs as performed by flow cytometry. A T-SNE analysis across all samples is shown to further investigate grouped effects. Lastly, the authors perform a PCR array of 84 genes and a gene expression analysis of SARS-CoV2 patients compared with healthy controls.  The main strengths of this paper are the even spread between male and female patients and being able to map the flow cytometry data back to patients.

The authors perform the flow cytometry experiments to a good standard and the indicated gating strategies were of high quality.  (Often there is not a live dead panel which is so important for this type of analysis).  It was unclear whether this was performed as a multiparameter panel or if some markers were performed individually.  Some of my specific comments relate to a potential reanalysis of flow data already performed if the data was acquired in a multiparametric fashion.

Main critique:

The authors set out to both phenotypically and functionally characterise the T cell populations derived from the patients of covid patients.  They have carefully characterised some of the phenotypic markers, but some of the claims focussing on exhaustion are too strong.  It is possible and highly likely that the difference seen in PD-1 expression between Covid patients and healthy donors is not exhaustion but just antigen recognition due to an active ongoing infection. Exhaustion should be sparingly used for those that expressed a range of markers that include PD-1, Tim-3, Lag3 CTLA4 etc and have functional anergy such as a reduction in target cell killing.

More specific questions/comments are below:

Figure 2:  To strengthen the claim of a numerical recovery rather than relative proportions of cells, a useful metric here would be to plot the specific numbers of CD4 and CD8 T cells pr ml of blood. CD4 and CD8 percentages are currently shown as a percentage of total cells and therefore, an increase in Neutrophils (shown) would have the effect of decreasing the overall percentage of T cells without reducing the true number.  Although I suspect the number has decreased as total Lymphocytes drop, as lymphocytes also contain other similar cells such as NK cells, NKT, MAIT and B cells, it is not possible to draw a conclusion that it is the loss of a/b T cells with the shown data.

Alternatively, a table showing all the lymphocyte counts from the blood would be sufficient to allow the reader to draw their own conclusions.

Figure 3: Were the CD38 and HLA-DR experiments performed as a multiparameter analysis?  If so could you please show the CD38/HLA-DR dual data.  Additionally, If performed as a multiparameter, it would be more useful to identify the CD8 and CD4 T cells that are CD38+, HLA+ and PD-1 Triple positive between patients.  Again, I would expect this to be the case in a virally infected individual as the majority of detectable cells would be antigen-experienced.

Small note: line 254 typo (MFI). 

Please include a line in the text as to the reason for including CD38 and HLA-DR as markers to ensure as wide an audience as possible can interpret the results. It was unclear why those markers were chosen over others. 

I would recommend the authors amend the text to reflect that PD-1 alone is not enough to establish exhaustion. PD-1 expression in critically ill patients could also indicate persistent infection and continued stimulation of T cells (Which of course could lead to exhaustion of these T cells, but at this stage the authors do not present the data to suggest this).

Figure 4:  I am not surprised that T cells from patients infected with COVID-19 are more readily producing IFN-g, Granzyme B and are able to degranulate at a basal level.  This likely reflects an active ongoing infection and that these T cells are effector T cells compared to (probably) Naïve T cells in the healthy donors.  Data not shown here include those with a moderate infection.  Was this performed? This would be a better control for comparison than healthy donor T cells.  Were the Healthy donor control cells activated with CD3/28 beads for 2 weeks prior to use in a cytotoxicity assay?

The CD8 data showing a decrease in perforin and CD107a (degranulation) but a huge increase in IFN-g over healthy controls is especially interesting and I feel more should be highlighted here.  Previous work by Jenkins et al (JexpMed 2015) has shown that perforin KO T cells unable to kill target cells show increased IFN-g release in Mice.  This may nicely correlate, and I’d invite the authors to take another look at that data.  This may also lead to potential cytokine storms seen in some patients and can lead to death.

Are you able to quantify the amount of cytokine that has been released through either an ELISA or cytokine bead array/Luminex assay?  This is important as currently the data just shows more cells are producing but they may have an impaired ability and therefore lower overall amount than 15-day activated healthy donor T cells.  

To properly determine the functional dysregulation of the CD8 T cells (and potential of cytotoxic CD4 T cells) I would recommend a redirected killing assay performed on the T cells from healthy Donors or severe/critical infection T cells.  An antihuman-CD3 antibody-coated target cell (P815 for example) in a simple flow-based assay over time and measuring target death should answer this question and strengthen any claims of impaired function, at least of direct cytotoxic T cell killing. Again I would include those of moderate infection as this is the better control as it eliminates the caveat of an ongoing infection being different to naïve T cell responses.

Figure 7:

Again I would include the data from moderate infection if possible.

If the data for moderate infection has not been performed then activation of healthy donor T cells over a period of 2 weeks to ensure a simulated Cytotoxic activation would be an adequate alternative.  It is unclear if this was what was performed or if the Healthy Donor T cells are collected straight from ficol/lymphoPrep preparation and stimulated with PMA. 

Author Response

Estimated reviewer,

First of all, I would like to thank you for the observations and suggestions for correction. All of them were very constructive and allowed for a better discussion and clarification of the manuscript.

All text changes are highlighted in yellow. Some graphics have been modified as suggested.

Below are the answers to the comments:

Figure 2: To strengthen the claim of a numerical recovery rather than relative proportions of cells, a useful metric here would be to plot the specific numbers of CD4 and CD8 T cells pr ml of blood. CD4 and CD8 percentages are currently shown as a percentage of total cells and therefore, an increase in Neutrophils (shown) would have the effect of decreasing the overall percentage of T cells without reducing the true number.  Although I suspect the number has decreased as total Lymphocytes drop, as lymphocytes also contain other similar cells such as NK cells, NKT, MAIT and B cells, it is not possible to draw a conclusion that it is the loss of a/b T cells with the shown data.

Reply:  We fully agree with this suggestion. We performed the calculations of CD4+ T lymphocytes and CD8+ T lymphocytes in absolute numbers (mm3), as shown in Figure 2 G and H. We observed a reduction in the absolute number of CD4+ T lymphocytes and CD8+ T lymphocytes, mainly in severe/critical patients.  Lymphopenia is evidenced in this metric (number of cells per mm3). Moreover, in Supplementary Figure 5, that shown the number of cells in different periods after positive PCR (there was no difference in numerical values, only lymphocyte recovery was observed after the first week of positive PCR in the frequency assessment). The text in the results section has also been modified to explain this new data.

Figure 3: Were the CD38 and HLA-DR experiments performed as a multiparameter analysis?  If so could you please show the CD38/HLA-DR dual data.  Additionally, If performed as a multiparameter, it would be more useful to identify the CD8 and CD4 T cells that are CD38+, HLA+ and PD-1 Triple positive between patients.  Again, I would expect this to be the case in a virally infected individual as the majority of detectable cells would be antigen-experienced.

Please include a line in the text as to the reason for including CD38 and HLA-DR as markers to ensure as wide an audience as possible can interpret the results. It was unclear why those markers were chosen over others. 

 I would recommend the authors amend the text to reflect that PD-1 alone is not enough to establish exhaustion. PD-1 expression in critically ill patients could also indicate persistent infection and continued stimulation of T cells (Which of course could lead to exhaustion of these T cells, but at this stage the authors do not present the data to suggest this).

 Reply: Thanks again for the remarks in Figure 3. We have corrected the indicated typo (MFI).

Unfortunately, we do not perform multiparametric analyses, which does not allow us to analyze double or triple positive cells.

As suggested, we discussed the reason for the analysis of HLA-DR and CD8+, making it clearer why these markers were chosen (lines 253-256 and lines 273-276).

Moreover, we amended the text (especially in the discussion) arguing that PD-1 expression in critically ill patients may also indicate persistent infection and continuous stimulation of T cells (lines 466-472 and 283-286).

 Figure 4:  I am not surprised that T cells from patients infected with COVID-19 are more readily producing IFN-g, Granzyme B and are able to degranulate at a basal level.  This likely reflects an active ongoing infection and that these T cells are effector T cells compared to (probably) Naïve T cells in the healthy donors.  Data not shown here include those with a moderate infection.  Was this performed? This would be a better control for comparison than healthy donor T cells.  Were the Healthy donor control cells activated with CD3/28 beads for 2 weeks prior to use in a cytotoxicity assay?

 To properly determine the functional dysregulation of the CD8 T cells (and potential of cytotoxic CD4 T cells) I would recommend a redirected killing assay performed on the T cells from healthy Donors or severe/critical infection T cells.  An antihuman-CD3 antibody-coated target cell (P815 for example) in a simple flow-based assay over time and measuring target death should answer this question and strengthen any claims of impaired function, at least of direct cytotoxic T cell killing. Again I would include those of moderate infection as this is the better control as it eliminates the caveat of an ongoing infection being different to naïve T cell responses.

 Figure 7:

Again I would include the data from moderate infection if possible.

 Reply:  We fully agree with the comments made. Indeed, lymphocytes from patients infected with SARS-CoV-2 are expected to produce greater amounts of cytotoxic factors. However, we draw attention to the difference in response between CD4+ and CD8+ T lymphocytes, even in the unstimulated situation.

Data do not include those with moderate infection. For phenotypic assays, we used EDTA whole blood samples from uninfected, moderate and severe/critical individuals. As for the functional assay and array PCR, heparin samples with greater blood volume were used to be able to separate PBMC and isolate CD8+ T lymphocytes. We had access to blood samples with heparin only from patients with severe/critical cases, as they remained hospitalized in the ICU unit, unlike moderate cases who did not remain in the hospital. We add that control cells were not activated with CD4/28 beads.

 We agree that should be important to analyze killing assay. However, we don’t have the samples used in this study were acquired at the beginning of the pandemic, before the beginning of the vaccination and infection protocols with the source strain of the virus. The blood samples volume was restricted, mainly for critical patients, not allowing to freeze the cells. Unfortunately, we no longer have access to these samples, under the same clinical conditions. As we know, the COVID-19 data now no longer represents the same clinical conditions at the beginning of the pandemic, which is when these trials were carried out.

Are you able to quantify the amount of cytokine that has been released through either an ELISA or cytokine bead array/Luminex assay?  This is important as currently the data just shows more cells are producing but they may have an impaired ability and therefore lower overall amount than 15-day activated healthy donor T cells.  

 Reply:Thanks for the experiment suggestions! Previously, we performed cytokine measurements by ELISA and this method was not able to detect sufficient amount of inflammatory cytokines (they were not detected in the patients' serum). Regarding performing a CBA, luminex, or redirected death test, we have an important limitation.

We have to inform you that the samples used in this study were acquired at the beginning of the pandemic, before the beginning of the vaccination and infection protocols with the source strain of the virus. Unfortunately, we no longer have access to these samples, under the same clinical conditions. As we know, the COVID-19 data now no longer represents the same clinical conditions at the beginning of the pandemic, which is when these trials were carried out.

Reviewer 2 Report

The manuscript by Gozzi-Silva, Sarah Cristina et al. describes the phenotype and function of T-lymphocytes in moderate and severe/critical cases of COVID-19. They showed an enhancement of CD4+ and CD8+ T-cytotoxic profiles in severe COVID-19 patients with more activated profile for CD8 compared to CD4 T cells. They conclude that dysfunctional cytotoxicity activity of CD8 T-cells was linked with the expression of genes associated with the caspase pathway as well as inflammasome showing to be more prone to death.

The study is clear, the paper is well written, and however I did not find the novelty, since 2020 many articles showed the phenotype and function of T lymphocytes.this is two examples:

doi.org/10.1002/eji.202049135 and doi :10.1038/s41467-020-17292-4.  

Many points have to be improved

-          You have to change the black color of the columns, we cannot visualize the points, you can use different grayscale, the same remark is valid for the hatched columns (figure 2G,H and I), please choose fine stripes to better visualize the points.

-          Explain the world (RNL) in Figure 2 and in the text for the first time you evoke it.

-          On the y-axis (Figure 2 B,C), please write simply % CD4+ T cells or CD8 T cells. The same for Figure 2 H and I.

-          Add please the + (% CD4+CD38+) in the legend of Figure 3A.

-          Please check if perforin data are not reversed between the two groups in Figure 4A and B, in the same figure, please restore the Y-axis, you have cut right where the error bar is, as the difference of the values is not big, I advise you not to cut the axis in two do not use the cuts of the axis. The same for supplementary figures.

-          Figure 4, you forgot to put the error bars for all the figures A-D.

-          Figure 6: Explain the worlds (TLR, NLR, RLR) in the figure legend and in the text.   

-          Please write Granzyme A in the supplementary figure 6 to don’t confuse with Granzyme B. In this figure the percentages of populations producing Granzyme A, IFNg or perforine are very low even after stimulation with PMA/Ionomycin, especially for CD4 T cells 0,25% or 1% is very weak percentages, CD8 T cells expressed always more cytotoxic molecules than CD4 T cells in this experiment. In this case, we cannot confirm that CD4 T cells can compensate the dysfunction of CD8 cytotoxic cells.

-          I have a doubt about the quality of same FACSs, for example, the percentage of PD-1 should be around 20% (more or less) especially in COVID patients (or other viral infections). But you show 0.5% for CD8 and 2% for CD4, which leaves a doubt either about the function of your antibody, or other technical problem? In this case, you could use other marker such as CD57, this is very important as the dysfunction/exhaustion of TCD8+ in acute SARS-CoV-2 infection is the main information of your study.

-          You talk about the expression of CD80 and TLR9 in the results of figure 6, but no explanation follows in the discussion, especially when we know that these two molecules are more expressed by APC rather than CD8

-          Differentially expressed genes in CD8+ T-lymphocytes of severe COVID-19 patients chow a decreased of TNF level (Figure 7M), however, TNF is increased by CD8 T cells from the same patients at protein level (Figure 4C), there is a contradiction, how do you explain it?

-          It would have been nice to measure some cytokines like TNF and cytotoxic molecules in the plasma of patients to confirm your observations.

-          Some in vitro stimulation experiments are necessary in this case. For example after CD4 and CD8 separation (ex vivo), in vitro cytotoxic assay to test the cytolytic capacity of T lymphocytes, theses in vitro data will reinforce your hypothesis.

Method section:

-          Protocol number for ethical committee missing  

-          Please add the concentration of anti CD107a that you put in the culture, in the section (Cytotoxic CD4+ and CD8+ T cells profile).

Author Response

Estimated reviewer,

First of all, I would like to thank you for the observations and suggestions for correction. All of them were very constructive and allowed for a better discussion and clarification of the present manuscript.

All text changes are highlighted in yellow. Some graphics have been modified for aesthetics as well as some graphics and analytics have been added as suggested

Below are the answers to the notes made:

You have to change the black color of the columns, we cannot visualize the points, you can use different grayscale, the same remark is valid for the hatched columns (figure 2G,H and I), please choose fine stripes to better visualize the points.

Reply: The colors of the graphics figures have been corrected to shades of gray (Figure 2, Figure 3 and Supplementary Figure 6) as well as thinner stripes have been added to the Figure 2 I-K.

Explain the world (RNL) in Figure 2 and in the text for the first time you evoke it.

 The word was included and explained - Neutrophil-Lymphocyte Ratio (NLR). Updated RNL to NLR which would be more suitable (line 382).

On the y-axis (Figure 2 B,C), please write simply % CD4+ T cells or CD8 T cells. The same for Figure 2 H and I.

Corrected the captions in Figure 2 for only %CD4+ and %CD8.

Add please the + (% CD4+CD38+) in the legend of Figure 3A.

Colors of the graphics in figure 3 have been corrected to grayscale and the legend of figure 3A has been corrected by adding the "+"

Please check if perforin data are not reversed between the two groups in Figure 4A and B, in the same figure, please restore the Y-axis, you have cut right where the error bar is, as the difference of the values is not big, I advise you not to cut the axis in two do not use the cuts of the axis. The same for supplementary figures.

As suggested, the y-axis section of figure 4 was removed. Perforin data are not inverted.

Figure 4, you forgot to put the error bars for all the figures A-D.

The error bar has been added in figure 4 as well as in the corresponding supplementary figure.

Figure 6: Explain the worlds (TLR, NLR, RLR) in the figure legend and in the text.   

The words indicated have been explained in the figure legend and in the text

 (line 382).

Please write Granzyme A in the supplementary figure 6 to don’t confuse with Granzyme B. In this figure the percentages of populations producing Granzyme A, IFNg or perforine are very low even after stimulation with PMA/Ionomycin, especially for CD4 T cells 0,25% or 1% is very weak percentages, CD8 T cells expressed always more cytotoxic molecules than CD4 T cells in this experiment. In this case, we cannot confirm that CD4 T cells can compensate the dysfunction of CD8 cytotoxic cells.

 Granzyme A was added. In fact, the production of granzyme and perforin are low especially in relation to CD4+ T lymphocytes. However, we draw attention that since the unstimulated situation, the CD4+ T lymphocytes of patients with COVID-19 show increased production of granzyme and CD107a in relation to the unstimulated situation of CD8+ T lymphocytes. This shows greater functionality of CD4+ T lymphocytes compared to CD8+ T lymphocytes in SARS-CoV-2 infection.

I have a doubt about the quality of same FACSs, for example, the percentage of PD-1 should be around 20% (more or less) especially in COVID patients (or other viral infections). But you show 0.5% for CD8 and 2% for CD4, which leaves a doubt either about the function of your antibody, or other technical problem? In this case, you could use other marker such as CD57, this is very important as the dysfunction/exhaustion of TCD8+ in acute SARS-CoV-2 infection is the main information of your study.

These data refer to the percentage of CD4+ and CD8+ T lymphocytes that express PD-1 within total living cells. For this reason, the percentage was lower. We make this information clearer in the figure legend.

You talk about the expression of CD80 and TLR9 in the results of figure 6, but no explanation follows in the discussion, especially when we know that these two molecules are more expressed by APC rather than CD8

In lines 525 a 532 the role of expression of these molecules in T lymphocytes was explained in greater detail.

Differentially expressed genes in CD8+ T-lymphocytes of severe COVID-19 patients chow a decreased of TNF level (Figure 7M), however, TNF is increased by CD8 T cells from the same patients at protein level (Figure 4C), there is a contradiction, how do you explain it?

In the lines 533-539, we explain the following to justify this data: “TNF is associated with the effector function of cytotoxic cells such as NK and CD8+ T-lymphocytes but also displays an immunosuppressive role, facilitating the biological activity of Tregs and myeloid-derived suppressor cells (Bertrand et al. 2016). The presence of TNF in CD8+ T-cells has a negative effect inducing the death of these cells, thus, the reduction of gene expression may indicate an attempt by CD8+ T-lymphocytes to maintain their effector function and prevent cell death”

It would have been nice to measure some cytokines like TNF and cytotoxic molecules in the plasma of patients to confirm your observations /  Some in vitro stimulation experiments are necessary in this case. For example after CD4 and CD8 separation (ex vivo), in vitro cytotoxic assay to test the cytolytic capacity of T lymphocytes, theses in vitro data will reinforce your hypothesis.

      We fully agree with the comments made and these assays would contribute to reinforce the cytolytic capacity of T lymphocytes. However, we have to inform you that the samples used in this study were acquired at the beginning of the pandemic, before the beginning of the vaccination and infection protocols with the strain virus source. Unfortunately, we no longer have access to these samples, under the same clinical conditions.

Method section:

Protocol number for ethical committee missing  

The ethics committee approval number has been included (line 112).

Please add the concentration of anti CD107a that you put in the culture, in the section (Cytotoxic CD4+ and CD8+ T cells profile).

The utilization factor used for the CD107a antibody was added (line 160).

Round 2

Reviewer 1 Report

Thank you for addressing my concerns and I take on board the technological limitations and access to samples due to the pandemic.  I'm happy with your changes and would just advise an extra read-through for English, but otherwise, I wish you all the best.

Author Response

We appreciate all your reviews and observations. English revisions were carried out.
We wish you all the best!

Reviewer 2 Report

Thank you to the authors for answering the questions, the answers were clear

Author Response

(The authors gave the same response as above.)
